# Biosensors: A Sneak Peek into Plant Cell’s Immunity

**DOI:** 10.3390/life11030209

**Published:** 2021-03-07

**Authors:** Valentina Levak, Tjaša Lukan, Kristina Gruden, Anna Coll

**Affiliations:** 1National Institute of Biology, Večna pot 111, 1000 Ljubljana, Slovenia; tjasa.lukan@nib.si (T.L.); kristina.gruden@nib.si (K.G.); anna.coll@nib.si (A.C.); 2Jožef Stefan International Postgraduate School, Jamova cesta 39, 1000 Ljubljana, Slovenia

**Keywords:** genetically encoded biosensors, live spatiotemporal imaging, crops, plant immune response, multiparameter imaging, biotic stress

## Abstract

Biosensors are indispensable tools to understand a plant’s immunity as its spatiotemporal dimension is key in withstanding complex plant immune signaling. The diversity of genetically encoded biosensors in plants is expanding, covering new analytes with ever higher sensitivity and robustness, but their assortment is limited in some respects, such as their use in following biotic stress response, employing more than one biosensor in the same chassis, and their implementation into crops. In this review, we focused on the available biosensors that encompass these aspects. We show that in vivo imaging of calcium and reactive oxygen species is satisfactorily covered with the available genetically encoded biosensors, while on the other hand they are still underrepresented when it comes to imaging of the main three hormonal players in the immune response: salicylic acid, ethylene and jasmonic acid. Following more than one analyte in the same chassis, upon one or more conditions, has so far been possible by using the most advanced genetically encoded biosensors in plants which allow the monitoring of calcium and the two main hormonal pathways involved in plant development, auxin and cytokinin. These kinds of biosensor are also the most evolved in crops. In the last section, we examine the challenges in the use of biosensors and demonstrate some strategies to overcome them.

## 1. Biosensors: Exploiting Molecular Hubs to Better Understand the Processes

A biosensor is a sensitive device that detects an analyte or event in a living organism and consequently produces a measurable output [1]. Thus, it consists of two functional parts: detector and reporter [2]. The detector domain must be specific and sensitive to enable detection of the biological concentration of analytes or events. The biosensor as a whole should enable detection with high signal to noise ratio (SNR), high spatiotemporal resolution on the organelle level, enable fast response, and must be functional in different cellular conditions (low pH, different redox states). Besides, it should not interfere with cellular processes and should not be toxic to the cells. Although each of available biosensors has its own limitations and cannot meet all of the abovementioned requirements, they are vital to obtain an insight into cellular events.

According to their mechanism of action, they can be separated into direct and indirect biosensors. Typical direct biosensors report protein activity upon ligand binding. Usually, they consist of two protein domains, one serving as a ligand receptor while the other can sense structural changes upon ligand binding and report it by measurable output. This is the mode of action of one of the first known biosensors named Cameleon, which is still widely used to measure Ca^2+^ concentration (Figure 1). It is translated as a single peptide chain, with two fluorescent proteins on each side, which are either blue and green or cyan and yellow, linked by calmodulin (CaM) and the M13 domain. When the concentration of Ca^2+^ in the cellular compartment increases, Ca^2+^ binds to CaM, which twists around the M13 domain and thereupon the two fluorescent proteins are close enough to enable Förster resonance energy transfer (FRET) and Ca^2+^ binding can be detected through the change of the fluorescence intensities [3]. Another kind of direct biosensors are degron-based biosensors, which undergo degradation as a result of analyte binding. They exploit the characteristics of the cellular signaling of some plant hormones. In the presence of the hormone in the cell, related transcriptional repressors are degraded and thus the transcription of the hormone-responsive genes is activated. This approach is used, for example, in a plant biosensor for auxin detection, known as DII-VENUS (Figure 1). This is a fusion protein of two domains, of which DII acts as a detector while fast-maturing fluorescent protein VENUS acts as a reporter. The DII domain is a part of the Aux/IAA repressor involved in the binding of auxin and subsequent degradation of the repressor via the ubiquitin/26S proteasome pathway. When auxin concentration in the cell increases, DII-VENUS fusion is degraded and the fluorescence intensity decreases [4]. Direct biosensors can be used for following changes in pH, redox state, ion and metabolite concentrations in the majority of plant cell compartments. 

Indirect biosensors are typically transcriptional reporters. Their detector domain is a promoter sequence that contains analyte-responding cis-elements and drives transcription of the reporter gene. The most commonly used reporters are beta-glucuronidase (GUS), fluorescent proteins (FPs) and luciferases. The signal produced by such biosensors is delayed, but amplified in comparison with direct biosensors. In order to gain higher SNR, specificity and sensitivity, native analyte-responsive promoter sequences are usually rebuilt by fusing cis-elements to the minimal Cauliflower mosaic virus (CaMV) 35S promoter sequence [5]. The most known example is the DR5 synthetic promoter, designed on the basis of GH3 gene promoter for auxin detection, which maintains its activity also in reverse orientation (DR5rev, Figure 1) [6]. The latter was improved to version DR5v2 with higher expression and sensitivity and therefore enables detection with better spatial resolution [7]. 

Recently, another type of indirect biosensor was developed to follow translational regulation of mRNA transcripts in the presence of ethylene. In this biosensor, the detector module is the ethylene-responsive 3’-UTR part of mRNA coding for EBF2, while the coding sequence is translated into reporter protein GFP (Figure 1). The mechanism of this biosensor is based on the action of C-terminal peptide of EIN2, which is cleaved when the concentration of ethylene increases. It can then bind to the 3’-UTR part from EBF2 mRNA fused with the GFP coding sequence, and thus represses its translation. In this way, GFP fluorescence decreases when the concentration of ET is increased [8,9,10].

During the last ten years, several informative reviews covering the topic of plant biosensors have been published, showing the advances in this technology and its importance for the plant research community. They revise the type of available biosensors and the recent results obtained [1], the use of biosensors to monitor plant hormones [11,12,13], the principles of most widely used biosensors in plants [14] and quantitative measurements with biosensors [15]. Some reviews are focused on biosensors of a single analyte such as abscisic acid (ABA) [16], auxin [17], Ca^2+^ [18], ethylene (ET) [10], gibberellins (GA) [19] and reactive oxygen species (ROS) [20]. The span of the developed biosensors goes hand in hand with advances of methodology, which exploits various principles and physical properties of the fluorescent proteins and other reporter proteins. The spectrum of advanced fluorescence imaging methods available nowadays for the use in plants is reviewed in Komis et al. [21]. The Fluorescent Biosensor Database [22] merges fluorescent genetically encoded biosensors regardless of the chassis and welcomes new updates by the community.

Variability of the biosensors available nowadays is broad. However, their use in plants is still limited to certain aspects. The majority of studies focus on the use of biosensors in roots to follow development in the model plant *Arabidopsis thaliana*. Consequently, the vast majority of biosensors were designed in this regard, providing even more than one sensor for a single analyte involved in a plant’s growth and development, each of them exploiting another cellular event. On the other hand, the span of available biosensors for immune response is narrow. While Ca^2+^ and ROS, involved in the first stages of signal transmission in general, are satisfactorily covered, the main hormones of immune response, salicylic acid (SA), jasmonic acid (JA) and ET, lack specific, sensitive and thus reliable biosensors to choose between. Leading research in plant development is also seen from the reports on following more than one analyte simultaneously and the transmission of the biosensors from model organisms to crops. Here, we report these findings. Finally, we also discuss difficulties associated with the application of the biosensors in plants with the aim of supporting the advancement of immune response biosensors and their application to crops.

## 2. Genetically Encoded Biosensors for Following Plant Immune Response 

Plants respond to biotic stress by reprogramming a complex signaling network that results in gene activity and metabolic changes. Translation of pathogen recognition into effective defense response strongly depends on the action of several plant hormones and other signaling molecules [23,24,25]. Early signaling events include changes of intracellular Ca^2+^ levels and a rapid increase of reactive oxygen species. Among plant hormones, SA, JA and ET have been identified as main players [24,26]. In addition, recent evidence shows that the effects of these three hormonal signaling pathways are balanced by ABA, GA, auxins, cytokinins and brassinosteroids (reviewed by Verma et al. [27]), thus adding another layer of regulation. Therefore, these are the best candidate analytes to follow general immune response (Figure 1). 

Ca^2^^+^ is the most versatile messenger regulating a wide range of responses, including biotic stress. After pathogen recognition, one of the earliest signaling events is the Ca^2+^ influx into the plant cell cytosol. Ca^2+^ signatures are also induced in the nucleus, mitochondria or chloroplasts and there is a complex interplay between Ca^2+^ and other messengers and their signaling pathways [38]. Moreover, it has been shown that Ca^2+^-mediated signaling is also involved in negative regulation of plant immunity [39]. Thus, there are still many questions that should be addressed. Ca^2+^ sensors are to date the most advanced among all biosensors. They have progressed from bioluminescent aequorin, which is natively anchoring calcium [40], to a wide range of designed fluorescent reporters employing calcium-binding domain CaM, e.g. FRET-based Cameleons (Figure 1) [3,41], single fluorophore GCaMPs (composed of circularly permutated enhanced GFP, CaM and M13 peptide) [42] and GECOs (genetically encoded Ca^2+^ indicators for optical imaging) [28] (Figure 1), and even GFP-aequorin, exploiting bioluminescence resonance energy transfer (BRET) [43]. Ca^2+^ response is fast and thus demands constitutively and strongly expressed direct biosensors, enabling subcellular [28], tissue and whole-plant imaging of calcium release reaching very high temporal resolution measured in seconds [44].

Despite the new insights that have been brought into the role of redox mechanisms in plant defence response, one of the major challenges still is to understand the spatial and temporal redox processes occurring during the defence response and to associate the transcriptional activity with the complex dynamics of these signaling molecules [45,46]. Several related biosensors have been successfully used in plants (reviewed in Choi et al. [47]) to help unravel the spatiotemporal redox signaling occurring during plant defense response. As an example, roGFPs are mutated GFP molecules sensitive to redox levels in the cell [48], which were later fused to signal sequences to allow targeting to different subcellular organelles, such as mitochondria [49] and chloroplasts [50]. Fusion partners known to be targets of redox transitions by glutathione (Figure 1) [30] or H_2_O_2_ (Figure 1) [29] were also designed and recently used for time-resolved measurements of both molecular species in the cytosol, chloroplasts and mitochondria [31]. This allowed for better understanding of the role of the mitochondria in sensing and signaling the cellular redox challenge in response to abiotic stress [51], deciphering the role of redox state in intercellular transport [50] or exploring the central role of glutathione in mediating redox signaling [52]. Another genetically encoded redox sensor used in plants is HyPer, based on circularly permutated YFP coupled with the H_2_O_2_-sensitive domain of OxyR, a transcription factor found in *Escherichia coli* [53]. A newly developed biosensor in plants, CROST (change in redox state of thioredoxin) employs a FRET-pair linked with redox-sensitive domain CP12 from *A. thaliana*, known to be reduced in vivo by thioredoxin [54]. Still, available H_2_O_2_ and redox sensors are frequently limited by extreme pH and redox conditions which must be considered when choosing the appropriate sensor for certain cellular compartments: apoplast, vacuole, endoplasmic reticulum, chloroplast, and mitochondria [20].

Recent advances in the development of plant biosensors (see review Novák et al. [11]) have helped to better understand dynamics of plant signaling, in particular in developmental processes [55]. Some of these can also be applied to monitor the immune response as some parts of the signaling network modules overlap. The most successful biosensors for hormones are transcriptional reporters, based on hormone-responsive promoter motifs fused to a reporter element. The first generation used the synthetic promoter fused to GUS, but, more recently, FPs have shown to be more versatile and have become the reporter of choice [56]. 

Transcriptional reporters have worked well not just in case of the abovementioned DR5 for auxin, but also in the case of the cytokinin Two Component System (TCS) [57] that was developed and used to uncover the roles of cytokinin signaling in *A. thaliana* root regeneration. It is named after the two-component phosphorelay cascade, the basis of the cytokinin signaling. The final target of the phosphorylation-triggered activation are transcription factors named B-type response regulators which activate the expression of cytokinin responsive genes. Their DNA binding sites are highly conserved and are thus exploited by TCS biosensors, joining six direct repeats to a minimal CaMV 35S promoter. The synthetic sensor TCS and its improved variants TCSnew (TCSn) [58] and TCS version 2 (TCSv2) [59] have been widely used in the model plant *A. thaliana*. Recently, a new synthetic transcriptional reporter for ABA was designed as six repeats of ABA-responsive elements (ABRE) from RD29A or ABI1 promoters fused to a minimal promoter, driving expression of either GUS or GFP targeted to endoplasmic reticulum. It was shown that both transcriptional reporters respond to osmotic stress [60].

Entanglement of the signaling pathways of the main three immune response hormones, SA, JA and ET, challenges the search for their specific transcriptional reporters. Promoters of pathogenesis-related proteins (PRs) are usually employed as transcriptional markers for SA signaling (Figure 1), while promoters of genes involved in JA biosynthesis (e.g. 12-oxo-phytodienoic acid reductase 3, OPR3), regulators of transcription (jasmonate-ZIM-domain 10, JAZ10) or target genes, e.g. plant defensin 1.2 (PDF1.2) and vegetative storage protein (VSP), are analyzed in connection with JA signaling (Figure 1). However, these promoters exhibit significant crosstalk between SA, JA and ET signaling pathways and are thus treated as defense-responsive genes. To our knowledge, specific genetically encoded promoter-based biosensors for SA and JA have not yet been developed in plants. ET transcriptional reporters are based on a synthetic promoter composed of five repeats of the Ethylene insensitive 3 (EIN3) binding site attached to the minimal CaMV 35S promoter [61]. So far, this has been used in combination with either luciferase or GUS due to its low strength [10]. Lately, some new promoters responding to ABA, auxin, cytokinin, SA and JA were identified in *A. thaliana* as promising candidates for transcriptional reporters [62]. 

Available direct biosensors that track plant hormone concentrations are based on either FRET or degrons. Recently, a major advance in high-resolution quantification of spatiotemporal GA distribution was achieved with the development of a sensor directly measuring GA. This exploits the interaction of GA receptor gibberellin insensitive dwarf 1 (GID1) with members of the DELLA family. Rizza et al. developed and implemented this FRET-based sensor (Gibberellin Perception Sensor 1, GPS1, Figure 1) in *A. thaliana* [34]. In fact, the first FRET-biosensors were developed for ABA measurement and were published in parallel by two different research groups: ABA concentration and uptake sensor (ABACUS) [63], and ABAleons [64].

Using a degron design, Larrieu et al. developed a biosensor for JA perception (Jas9-VENUS, Figure 1) and demonstrated its value for quantitative and dynamic analysis of JA response in *A. thaliana* roots [65]. The Jas9-VENUS biosensor uses the Jas motif of Jasmonate-ZIM-Domain (JAZ) proteins that are targeted to degradation via the ubiquitin/26S proteasome pathway in the presence of the bioactive form of JA. Another degron-based biosensor designed to monitor GA is the GFP-tagged DELLA protein repressor of GA1-3 (GFP-RGA, Figure 1) that was used in the model plant *A. thaliana* and revealed asymmetric distribution of GA and GA signaling during root gravitropic growth [33]. StrigoQuant is also a degron-based but luminescent reporter which includes two luciferases, firefly and Renilla luciferase (FLUC and RLUC, respectively) that are expressed under the same promoter. Co-translationally, fusion of FLUC and RLUC is cleaved by 2A self-cleaving peptide which links both reporters: one is degraded upon presence of strigolactons, the other is used for normalization of expression [66].

Both direct and indirect biosensors are now frequently designed along with their nonresponsive counterparts, which were constructed in the same way but are not responsive to the analyte. Thus, they show the background signal from the biosensor itself, and can be used directly for normalization when fused to another reporter protein. Some examples are nlsGPS1 and nlsGPS1-NR (gibberellin) [67], Jas9-VENUS and mJas9-VENUS [65], DII and mDII (e.g. R2D2: DII-3xVENUS and mDII-ntdTomato) [7], TCS and TCSm [57].

Other useful biosensors of immune response are those following the activity of mitogen-activated protein kinases (MAPKs) through the use of docking domain and phosphorylation site (FRET biosensors and KTRs, kinase translocation reporters, in which a reporter changes its intracellular localization when phosphorylated), namely *A. thaliana*’s MPK3, MPK4, MPK6, which take part in response to flg22, chitin and NaCl (Figure 1) [36,68]. The SnRK2 (sucrose nonfermenting-1-related kinase 2) activity sensor (SNACS) is another kinase activity FRET biosensor for ABA-responsive SnRK2 protein kinases involved in stomata closure, stably transformed in *A. thaliana* and its mutants [69].

To follow the outcome of defence and growth antagonism in the whole plant, it is possible to engage cell division reporters based on cyclins, for example, CYCB1;1, B-type cyclin that is present only in late G2 phase and early M phase of the cell cycle: AtpCYCB1;1::CYCB1;1-tYFPnls (Figure 1), also available in fusion with GUS [35], or CYCD6;1, driving S-phase of DNA replication: CYCD6;1::GFP [70]. Recently, a three-component biosensor was designed to follow the whole cell cycle in *A. thaliana* [71]. It contains a multigene cassette expressing fusions of CDT1a-eCFP, HTR13-mCherry and N-CYCB1;1-YFP, each under the control of the native promoter of the first fusion partner. CDT1a is involved in initiation of DNA replication and is expressed during the S and G2 phases, HTR13 is histone H3.1 protein and is expressed during the M and G1 phases, while N-CYCB1;1 is expressed during the late G2 and prophase and metaphase of mitosis [71]. Some possible solutions have not yet been applied to plants. Recently, another promising type of biosensor was established in the animalia kingdom, FlipGFP, for following protease activity. This is based on tripartite GFP, which fluoresces only when reconstituted with two missing beta-strands. These become available after the cleavage of their fusion by a specific protease [72].

## 3. Getting a Broader View and Deeper Understanding: More Than One Biosensor in the Same Chassis

In the field, plants are rarely exposed to a single stress. They often interact with different organisms, either simultaneously or sequentially, and can be affected by several abiotic stresses such as drought, heat or salinity. The effects of these adverse conditions in plant growth and yield can be devastating and are becoming more problematic with the advent of climate change. Understanding the response of plants to environmental conditions and how the different signaling pathways interact is crucial to guarantee efficient crop protection strategies. Several studies have reported that the involvement of different signaling pathways in response to multiple stresses and three-way interactions cannot be inferred from the response to a single stress [73,74]. Therefore, it is of high importance to gain better insights into the plant response to multiple stresses and the use of combined biosensors could be a promising tool to greatly advance this field.

Employing more than one biosensor simultaneously can provide information for more than one analyte or for the same analyte in more than one cell compartment. Although it demands additional efforts in cloning, transforming, imaging and analysis, examples have recently shown the value of this approach. Waadt et al. [75] investigated the interdependence of calcium and ABA signaling in *A. thaliana* roots. The authors performed multiparameter imaging of both analytes combining the red-emitting single-FP genetically encoded Ca^2+^ indicators for optical imaging (R-GECO1) [76] and the FRET-based ABA reporter ABAleon2.1 emitting in cyan/yellow [64]. Taking advantage of the high sensitivity of GECOs, a dual sensor for monitoring Ca^2+^ signal dynamics in the cytoplasm and the nuclear compartments was developed by assembling the nuclear-R-GECO1 (NR-GECO1) and the cytoplasmic green GECO1 (CG-GECO1) in a single construct [28]. The dual GECO sensor was shown to be a useful tool to monitor Ca^2+^ signal response to biotic and abiotic stress of *Medicago truncatula* and *A. thaliana* roots [28]. Another example of dual sensors is the use of transgenic *A. thaliana* plants simultaneously expressing DR5::3xVENUS-N7 [77] and TCS::GFP [57] reporters to study the spatial patterns of auxin and cytokinins, respectively [78].

The analysis of several analytes simultaneously requires the generation of transgenic plants that express several genetically encoded sensors by co-transformations with single transcriptional units or transformation with a multigene cassette following the assembly of single sensors. The generation of transgenic plants is time-consuming and the insertion of multiple transgenes into the *A. thaliana* genome could result in epigenetic silencing effects [79]. In contrast, it has been shown that the inducible co-expression of two interacting proteins, each tagged with FP, in a single multigene expression cassette reduces variability in expression of the proteins in a single cell, avoids mosaicism and can increase FRET [80]. By expressing two biosensors from one single mRNA Waadt et al. introduced the concept of dual-reporting transcriptionally linked genetically encoded fluorescent indicators (2-in-1-GEFIs). The two fluorescent proteins are separated by 2A self-cleaving peptide. This sensor was used for the multiparametric analysis of ABA, Ca^2+^, protons, chloride, the glutathione redox potential, and H_2_O_2_ in *A. thaliana* roots [81].

Many of these approaches are not specific to fluorescent proteins, but can be also applied to the constructs with luciferases. Recent research provided new luciferases that emit light of different wavelengths and can thus be used simultaneously (similar to FPs), or even in fusion with FPs in BRET experiments [82]. Apart from StrigoQuant [66], a pair of FLUC and RLUC was used as a sensor for miRNA silencing, where FLUC transcript was miRNA’s target while RLUC was used for normalization [83]. More recent luciferases that complement each other with respect to emission wavelength are the red firefly luciferase (redLUC) and Gaussia Dura luciferase (gLUC) that have already been used in a similar manner to StrigoQuant [84]. NanoLUC is smaller than other luciferases, does not use ATP and allows higher temporal resolution [85]. Light produced from another luciferase, GeNL, performs high transmittance through plant tissue when used with appropriate substrate [86].

In order to follow various molecules and more aspects of a response in a spatiotemporal manner, different approaches can be used simultaneously. We can exploit the options of non-genetically encoded biosensors such as nanosensors which are applied onto the surface, such as carbon nanotubes that change their fluorescence when exposed to higher H_2_O_2_ concentrations [87]. It has also been shown in *A. thaliana*, wheat and maize that plants can be fumigated with permeable fluorescent probes that irreversibly react with ROS [88]. Microbial biosensors, such as *Acinetobacter* sp. ADPWH_*lux*, can use SA as a sole carbon source and can therefore be exploited for SA detection (Figure 1) [32]. Such an approach is especially useful to gain appropriate time resolution when we want to avoid laborious and time-consuming stable transformation or when stably and constitutively expressed reporters cannot be obtained.

Single or multiple sensors can be used in parallel with reporter microorganisms that allow monitoring of the spatiotemporal response over the plant in relation to the signal from the genetically encoded sensor. Among these we can find GFP-coding viral genome or GFP-tagged infectious viral particles such as plum pox virus [89], potato virus X [90], cowpea mosaic virus [91] or potato virus Y [92]. Several FP-tagged parasitic and (endo)symbiotic bacteria have also been developed, while GUS reporter must be used cautiously because some microorganisms show strong GUS or GUS-like background activity [93,94]. However, the use of lacZ-labelled *Rhizobium leguminosarum* to study infection and nodule development in legumes [95] and the efficiency of luminescent *Ralstonia solanacearum* reporter [96] as a tool to assist potato breeding programs have been recently published. In the case of bacteria, an improved variant of self-assembling split super-folder green fluorescent protein system was optimized to investigate the spatiotemporal dynamics of effectors delivered by the bacterial type III secretion system into the plant cells [97]. Moreover, some FP-tagged fungi are also available, such as *Magnaporthe oryzae* [98,99], *Fusarium graminearum* [100] *and Fusarium solani* [101]. The availability of reporter-incorporated microorganisms is useful for studying biofilm formation, to follow plant-host interaction and, in case of fungi, it enables imaging of hyphae formation, from its passage through the apoplast to its entry into plant cells. Combined with multiple genetically encoded sensors, this approach would allow us to follow the plant response at the site of the interaction with spatiotemporal resolution.

## 4. Responses of the Cultivated: Biosensors in Crops

While *A. thaliana* serves as the playground where functionality of biosensors developed for animalia kingdom are usually first used, crops seem left behind, as only the most used examples reach them after a delay. The reason for this frequently lies in unsuccessful attempts to produce functionally stable transformants. Stable transformations of some plant species are demanding due to the larger genome and higher ploidy, higher content of repetitive regions and topologically associated domains (TADs), local intra-chromosomal contacts that are species-specific [102]. Added to the challenge of transforming crops is the fact that most can be unresponsive to tissue culture protocols and they have longer growing seasons. Some crops lack (fertile) seed production which also makes it harder to produce crossed lines that contain two or more stacked transgenes. Consequently, the analyses of responses of crops to various stimuli mostly depend on transcriptomics studies while the confirmation of results using biosensors is limited to the transient transformation of *Nicotiana* spp. or *A. thaliana* protoplasts, which can provide much faster but less accurate results. However, this restricts the span of the observable genes to those that have orthologues in further-related species while those with different functions often fail to be more closely observed in their native species. 

Despite the limitations, there are some crops available with biosensory properties (Table 1). Ideally, the use of stable transformants is the most desirable approach but it is time consuming and frequently unsuccessful. In some species, mainly in the family of the legumes (but also many other species, e.g. barley [103]), *Agrobacterium rhizogenes*-mediated transformation of roots provides an alternative approach [104]. These can suffice for following various aspects of root development, interaction with microbiota and nodulation.

Usually, the sensors that have been applied to crops are the widely used transcriptional reporters for cytokinin or auxin responses consisting of a synthetic promoter and a reporter. In contrast, from the wide range of remaining available sensors, to our knowledge, only the ones that enable monitoring Ca^2+^, ROS and cell division have been applied to a reduced number of plant species other than *A. thaliana* (Table 1). 

Interestingly, multiparameter imaging has already been used in roots of legumes to follow relations between auxin and cytokinin signaling during root growth and nodulation. Fisher et al. assembled a multigene cassette carrying the GFP under the control of synthetic promoter DR5 and the tandem-dimer Tomato (tdTomato) under the control of the TCSn promoter, which enabled them to determine auxin and cytokinin response and their ratios in root and nodule tissues of soybean [125]. Similarly, Nadzieja et al. monitored auxin and cytokinin response of *Lotus japonica* roots, transformed with DR5::mCherry-NLS and TCSn::YFP-NLS sensors, expressed from the same multigene cassette [35]. To detect the spatial correlation between inoculation with symbiotic bacteria *Mesorhizobium loti* and cytokinin or auxin response, DsRed-marked bacteria were applied to roots of *L. japonicus* expressing either TCSn::YFP-NLS [131] or DR5::GUS [116] biosensor, respectively. DR5::tdTomato biosensor was co-expressed in soybean with GFP under constitutive promoter super ubiquitin (sUbi::GFP). Spatial overlap of the signals from FPs enabled discrimination between the red signal from the sensor and bright red autofluorescence [126] (Table 1). 

## 5. Challenges of Plant Biosensors’ Development and Use 

There are many successful reported uses of biosensors, though their development can be challenging and obtained results are not always straightforward to interpret. While the vast majority of biosensor detection has been performed in plant roots, protoplasts and transiently transformed tobacco pavement cells, imaging of photosynthetic tissue is far less commonly reported. It is often limited to whole plant imaging which does not allow high spatial resolution, while leaf tissue close-ups with subcellular resolution are rare. Besides some general instrumentation constraints, for instance focus drift and unstable laser power in the first couple of hours, uneven illumination and time-consuming high-quality image acquisition [134], imaging of FPs in plants is affected by fluorescent compounds found in cuticle, cell walls, plastids and vacuoles, such as lignin, chlorophyll and other pigments, etc. [135]. These components generate high background and cause low SNR in detection of fluorescent reporters. Apart from that, they can mislead our interpretation of subcellular organization and structures. For example, during their imaging of GFP-tagged plasmodesmata, Liu et al. reported strong reflection from the cell wall, which could be mistakenly interpreted as plasmodesmata structure in both of the neighboring cells [136]. They were able to avoid misinterpretation due to gene gun transformation with lower density of transformed cells compared to agrobacteria-mediated transformation. As microscopy is very time consuming it is in some cases not the method of choice, especially in the first phases of biosensor development, such as optimization of promoter sequences. Microplate reader fluorimetry/luminometry enables fast screening of many biological replicates simultaneously and detection of crucial time points for further more detailed observation [137,138], but lacks spatial resolution. 

To follow a plant’s dynamic response through space and time with high resolution, the problem of SNR should be addressed. Partly, we can achieve optimal SNR with proper image acquisition and processing, and also region of interest (ROI) selection [134,139], but this is not always sufficient. There are different strategies to overcome the problem of low SNR at the level of biosensor construction. One option aims to overpower autofluorescence with high expression of fluorescent proteins. This is commonly achieved with the help of minimal CaMV 35S promoter. For transcriptional reporters the promoter can be coupled with CaMV 35S enhancer region and two or more repeats of inducible parts of a certain promoter, e.g. PR2 (from parsley) or AtCMPG1 [140,141]. Higher expression can also be achieved with the use of proper combinations of 5’-UTR and 3’-UTR enhancer regions [142], compatible with the plant of choice [143] as it is done for protein production in planta. When a transcriptional reporter is under the control of a weak promoter, an additional transcriptional regulator between the two can amplify its activity [144]. However, this effort is frequently opposed by silencing [137]. 

The proper choice of localization can also heighten SNR. The signal in plant cytoplasm is weak, variable among cells, hard to quantify and more prone to silencing. On the other hand, localization in other compartments can also have some constraints. Reporter protein localized in vacuole and apoplast should be pH stable, a condition refusing many widely used fluorescent proteins [145]. For this reason, red variants of redox sensors roGFP and HyPer were designed, namely Grx1-roCherry [146] and HyPerRed [147], respectively. Additionally, HyPer and HyPerRed have their redox-non-sensitive and pH-sensitive counterparts and can be used in parallel as a control of pH effect on measurements [147]. However, we have not found any report on their use in plants so far. Nucleus and mitochondrial localizations also have drawbacks. For example, small fluorescent proteins can exhibit leaking when tagged with either nuclear localization signal (NLS) or nuclear exportation signal (NES), while the mutants expressing fluorescent biosensors in the mitochondrial matrix were shown to grow more slowly than others [29]. 

Higher analyte specificity is also helpful when dealing with low SNR. This can be reached with a chimeric effector/detector module [148]. This often comes with lower constant of dissociation for the chosen analyte and can therefore interfere with the analyte’s availability for endogenous targets, which was the case in ABA sensors (reviewed in Isoda et al. [13]). Potential slow release of the analyte from the binding pocket of direct biosensors must also be considered to avoid misinterpretations of temporal dimension of the analyte availability in vivo [67]. 

Lower SNR in transformants after agrobacteria-mediated transient transformations can be caused by the expression of reporter proteins in agrobacteria, which can be overcome by the promoter exchange [149] or the insertion of an intron [150], the last resulting in higher expression in plant cells through intron-mediated enhancement [151].

Another common problem affecting the use of biosensors is their capacity for analyte quantification. Cell responses are not binary, but rather pattern- and concentration-dependent, so the need for biosensors enabling dynamic quantitative imaging arises, especially when high spatiotemporal resolution and quantitative data are needed for systems biology approaches [152]. Aequorin is a perfect example of absolute intensiometric biosensor for Ca^2+^ imaging as it enables measurement of absolute in vivo concentration [153]. Non-FRET ratiometric biosensors use a duet of reporter proteins, one as a reporter of analyte and another as a reporter of expression in a certain cell or tissue. The second can be expressed separately as a normalization transcriptional unit under constitutive promoter. Expression under strong viral constitutive promoter CaMV 35S can experience silencing or patterns in dividing cells [7] and is therefore often exchanged with plant constitutive promoters, such as rice actin and maize ubiquitin promoters. Still, variable expression in different species forces the exploration of novel options [154,155]. On the other hand, it is possible to opt for separation of two or more reporters at the protein stage. One possibility is co-translational separation by a self-cleaving peptide [84,156]. However, after the cleavage, the peptide is not excised but remains attached to the C-terminus of the upstream protein sequence which can affect its folding or function. This can be overcome by attachment of a peptide linker, which is a target of endogenous peptidases [157,158], or mini-intein with N-terminal autocleavage ability [159]. Synchronized expression of two proteins can also be obtained by using a polyprotein vector system that is based on a pair of self-excising mini-inteins, called dual-intein domain, which allow the release of both proteins (shown for tripartite sfGFP) [160]. In contrast, the use of internal ribosomal entry site (IRES) was not successful and is not recommended, as the level of IRES-driven translation can vary among cells [160]. 

To alleviate the problems associated with fluorescence imaging in plants, one can also use luciferases as the reporter domain in biosensor. One of the drawbacks of luciferases is the need for external application of the substrate which might not readily penetrate into plant tissue. To overcome this issue, autoluminescent *N. benthamiana* plants were engineered by the insertion of a fungal bioluminescence gene cluster (all with CaMV 35S promoter) [161]. Thus, the transgenic plants do not need any external substrate addition, as they produce fungal luciferin from caffeic acid. Treatment with methyl jasmonate, ethylene and wounding caused higher luminescence within seconds [161,162]. However, the use of the plant’s metabolite impacts the final luminescence produced according to availability. For example, older leaves showed lower luminescence [161,162]. A similar approach, avoiding exogenous substrate application, was explored in the reporter RUBY which produces red betalain pigment. The three enzymes that cooperate in its biosynthetic pathway from tyrosine were expressed under the control of various promoters in *A. thaliana* and were co-translationally separated due to the addition of 2A self-cleaving peptide [163].

## 6. Concluding Remarks

Biosensors have become indispensable tools to gain new insights in molecular biology with high spatiotemporal resolution. When being transferred to plants, especially crops, the community experiences challenges. However, overcoming these challenges is more and more supported by new achievements in synthetic biology, imaging and plant transformation fields, and can lead to new discoveries. Biosensors have so far been used individually, tracking only one analyte per experiment, with rare exceptions. We believe that the field of biosensors is now ready for multiparameter imaging. Therefore, this approach should now be used to obtain (quantitative) data with high spatiotemporal resolution that offer high-quality input for mathematical modeling of the dynamic network of plant responses to environment. Biosensors are promising tools to uncover the mysteries of a plant’s orchestrated signaling network that leads to discrimination between specific immune responses. 

## Figures and Tables

**Figure 1 life-11-00209-f001:**
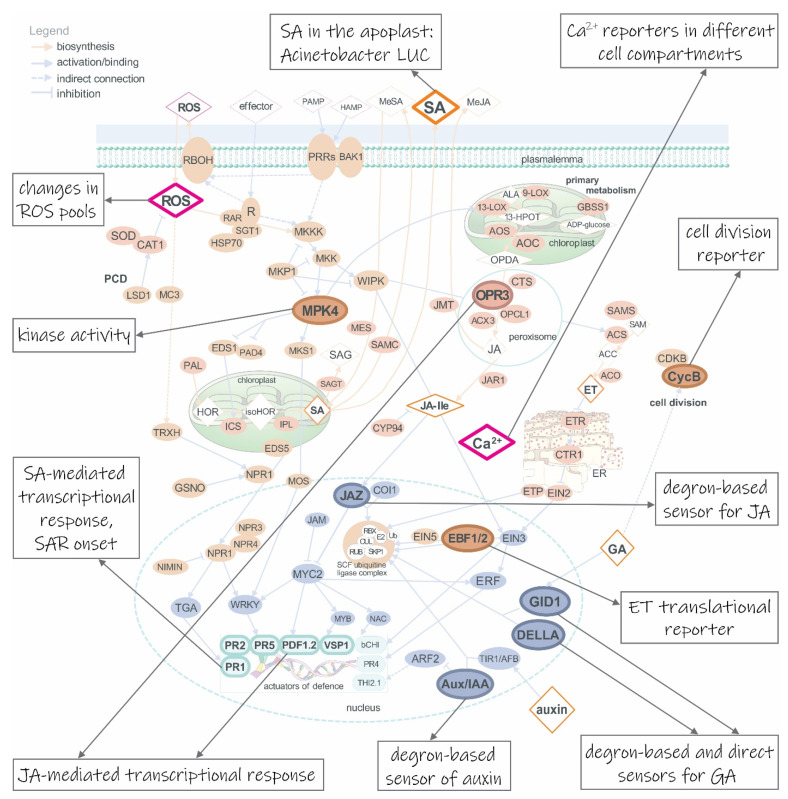
Diversity of available biosensors for following immune response in plants. During the first stage of the immune response, Ca^2+^ influx in the cytosol can be monitored with diverse reporter proteins. They can be Förster resonance energy transfer (FRET) based, employing two FPs linked by calmodulin-sensing domain, e.g. Cameleons [3], or colourful GECOs (genetically encoded Ca^2+^ indicators for optical imaging) [28], employing one, circularly permutated fluorescent protein. Prominent sensors of reactive oxygen species (ROS) are redox-sensitive GFPs in fusion with oxidant receptor peroxidase-1 (roGFP2-Orp1) [29] and glutaredoxin 1 (Grx1-roGFP2) [30], H_2_O_2_ and glutathione sensors, respectively, that were used to distinguish the patterns of these two molecular species in the cytosol, chloroplast and mitochondria upon illumination [31]. The main three hormones of immune response, salicylic acid (SA), jasmonic acid (JA) and ethylene (ET), can be followed with a direct degron-based sensor Jas9-VENUS in case of JA or with indirect transcriptional reporters based on defense genes’ promoters, such as pathogen-responsive PR1, PR2 and PR5 for SA- and plant defensin 1.2 (PDF1.2) and vegetative storage protein (VSP) for JA-involving response. They can be additionally followed by transcriptional reporters that exploit promoters of genes involved in hormone biosynthesis, e.g. the promoter of 12-oxo-phytodienoic acid reductase 3 (OPR3) gene, a constituent of JA biosynthesis. However, these transcriptional reporters can only complement other biosensors as they are not sufficiently specific. SA in the apoplast can be followed through *Acinetobacter* sp. ADPWH_*lux* luciferase (LUC) activity [32]. ET presence can be followed with translational reporter, joining 3’-UTR mRNA of EBF2, repressor of ET-responsive genes, and coding sequence of FP [8,9]. Other hormones, such as auxin and gibberellins (GA), can also be followed with degron-based sensors, employing DII domain of Aux/IAA repressor (in case of auxin [4]) and DELLA repressor (in case of GA [33]). Auxin can also be monitored through more of its actions, not just directly with DII-FP, but also indirectly with transcriptional reporters of auxin-responsive genes. These can employ synthetic (DR5 [6]) promoter as a detector. On the other hand, GA can be followed with FRET-based direct biosensor exploiting interaction between GID1 and DELLA repressor [34]. Interplay of immune response and plant’s growth and development can be followed with biosensors of cell division, employing cyclins, e.g. CycB [35]. AtMPK4 and some other kinases’ activity can be detected with kinase localization reporters, employing a domain for kinase docking and a phosphorylation site. When phosphorylated, fluorescent reporter is exported from the nucleus [36]. Signaling proteins are presented as orange, enzymes as brown-red and transcription factors as grey ovals, respectively; metabolites are presented as rhombs and genes as squircles. Full and dotted arrows represent direct and indirect connection; orange and grey arrows represent biosynthetic pathways and binding; point and block arrows represent activation and inhibition, respectively. ARF2, auxin response factor 2; Aux/IAA, a member from the protein family of auxin-sensitive transcriptional repressors; CDPK, cyclin dependent phosphokinase; DELLA, a member of the protein family of GA-sensitive transcriptional repressors; GID1, gibberellin insensitive dwarf 1; JAZ, jasmonate ZIM-domain; MPK4, mitogen-activated protein kinase 4; PR1, 2 and 5, pathogenesis-related protein 1, 2 and 5, respectively. Figure was adapted on the basis of Figure 1c from Lukan et al. [37] with author’s permission.

**Table 1 life-11-00209-t001:** Examples of genetically encoded biosensors applied to crops. Comments are added to those biosensors that were used in multiparameter imaging.

Analyte	Biosensor	Crop	Transformation	Comments	Reference
Ca^2+^	NES-YC3.6	*Lotus japonicus*	stable		[105]
	NLS-YC3.6	*L. japonicus*	stable		[105]
	NRCG-GECO1	*Medicago* *truncatula*	roots - transient	dual sensor localized in nucleus and cytoplasm	[28]
	NupYC2.1	*M. truncatula*	roots - transient		[106]
	aequorin	potato	stable		[107]
	aequorin	tomato	stable		[108]
	YC3.6	tomato	stable		[109]
ROS	roGFP1	tomato	stable		[110]
ROS: GSH	chl-roGFP2	potato	stable		[111]
ROS: H_2_O_2_	HyPer	*M. truncatula*	roots - transient		[112]
auxin	DII-VENUS	*Brachypodium* *distachyon*	stable		[113]
	DR5::nlsGFP	*Hieracium* *piloselloides*	stable		[114]
	DR5::GFP-NLS	*L. japonicus*	roots - transient		[115]
	DR5::GUS	*L. japonicus*	roots - transient	inoculated with DsRed-tagged rhizobium	[116]
	DR5::mCherry-NLS	*L. japonicus*	roots - transient	co-expressed with TCSn::YFP-NLS	[35]
	DR5::tYFPnls	*L. japonicus*	roots - transient		[35]
	R2D2	*L. japonicus*	roots - transient	co-expression of DII-tYFPnls and mDII-NLS-DsRed	[35]
	DII-VENUS-NLS	maize	stable		[117]
	DR5rev::mRFPer	maize	stable		[118]
	DR5::GUS	*M. truncatula*	stable		[119]
	DR5::VENUS-N7	*M. truncatula*	stable		[120]
	DR5rev::GFP	*M. truncatula*	stable		[119]
	DR5::GUS	poplar	stable		[121]
	DR5rev::3XVENUS-N7	potato	stable		[122]
	DR5rev::3xVENUS-N7	rice	stable		[123]
	DII-VENUS	rice	stable		[123]
	DR5::GUS	*Senecio* *vulgaris*	stable		[124]
	DR5::GFP-NLS	soybean	roots - transient	co-expressed with TCSn::tdTomato-NLS	[125]
	DR5::tdTomato	soybean	roots - transient	co-expressed with sUbi::GFP	[126]
	DR5::GUS	tomato	stable		[127]
	DR5rev::3xVENUS-N7	tomato	stable		[128]
	DR5rev::mRFPer	tomato	stable		[129]
cytokinin	TCSn::VENUS-H2B	barley	roots - transient		[130]
	TCSn::YFP-NLS	*L. japonicus*	roots -transient	co-expressed with DR5::mCherry-NLS	[35]
	TCSn::YFP-NLS	*L. japonicus*	stable	inoculated with DsRed-tagged rhizobium	[131]
	TCSn::GUS	rice	stable		[132]
	TCSn::tdTomato-NLS	soybean	roots - transient	co-expressed with DR5::GFP-NLS	[125]
	TCSv2::3xVENUS	tomato	stable		[133]
	TCSv2::GUS	tomato	stable		[133]
cell division	CycB1;1::GUS	potato	stable		[122]

NES- and NLS-YC3.6, yellow cameleon 3.6 with nuclear export and nuclear localization signal, respectively; NRCG-GECO1, nucleus-red, cytosol-green GECO1; NupYC2.1, nucleoplasmin-tagged yellow cameleon 2.1; tYFPnls, triple repeat YFP with nuclear localization signal; R2D2, biosensor composed of DII-3xVENUS (fusion of DII and triple repeat VENUS yellow fluorescent protein) and mDII-ntdTomato (fusion of mDII and nuclear tandem dimer Tomato red fluorescent protein); mRFPer, monomeric RFP targeted to endoplasmic reticulum; N7 and H2B, nuclear localization signals, derived from ankyrin-like protein and histone H2B.

## Data Availability

Not applicable.

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
