# Peer review of "Biosensors: A Sneak Peek into Plant Cell’s Immunity"

_life, 2021, doi:10.3390/life11030209_

Round 1

Reviewer 1 Report

Please find my comments and suggestions in the pdf document. Thanks.

Reviewer 2 Report

In the manuscript ‘Biosensors: a sneak peek into plant cell’s immunity’, Levak and collaborators reviewed the knowledge on biosensors developed to follow plant immune response, in particular to monitor calcium, ROS and hormones.

The manuscript is extraordinarily well written and well organized. The review has high significance and it could prove to be very interesting and useful to a very large audience. The review gives the reader an overview of the wide range of applications of biosensors in plants. The authors not only make a revision of knowledge but also point out possible solutions to be applied to plants based on the findings obtained in animal research, which is very good. Just one specific comment: check line 329 (such as).
